# Self-Assembled BODIPY Nanoparticles for Near-Infrared Fluorescence Bioimaging

**DOI:** 10.3390/molecules28072997

**Published:** 2023-03-28

**Authors:** Jiale Wang, Zhao Jiang, Cheng Huang, Shimao Zhao, Senqiang Zhu, Rui Liu, Hongjun Zhu

**Affiliations:** 1School of Chemistry and Molecular Engineering, Nanjing Tech University, Nanjing 211816, China; 2Department of Chemical and Pharmaceutical Engineering, Southeast University ChengXian College, Nanjing 210088, China

**Keywords:** BODIPY, self-assembly, near-infrared, fluorescence property, bioimaging

## Abstract

In vivo optical imaging is an important application value in disease diagnosis. However, near-infrared nanoprobes with excellent luminescent properties are still scarce. Herein, two boron–dipyrromethene (BODIPY) molecules (BDP-A and BDP-B) were designed and synthesized. The BODIPY emission was tuned to the near-infrared (NIR) region by regulating the electron-donating ability of the substituents on its core structure. In addition, the introduction of polyethylene glycol (PEG) chains on BODIPY enabled the formation of self-assembled nanoparticles (NPs) to form optical nanoprobes. The self-assembled BODIPY NPs present several advantages, including NIR emission, large Stokes shifts, and high fluorescence quantum efficiency, which can increase water dispersibility and signal-to-noise ratio to decrease the interference by the biological background fluorescence. The in vitro studies revealed that these NPs can enter tumor cells and illuminate the cytoplasm through fluorescence imaging. Then, BDP-B NPs were selected for use in vivo imaging due to their unique NIR emission. BDP-B was enriched in the tumor and effectively illuminated it via an enhanced penetrability and retention effect (EPR) after being injected into the tail vein of mice. The organic nanoparticles were metabolized through the liver and kidney. Thus, the BODIPY-based nanomicelles with NIR fluorescence emission provide an effective research basis for the development of optical nanoprobes in vivo.

## 1. Introduction

Bioimaging is an important tool used for resolving the structure of biological tissues and elucidating physiological processes [1,2,3]. By adopting a particular monitoring approach to the cells and tissues in an organism, it is possible to detect specific changes and thus visualize and quantify the biological, physiological, and pathological processes on a cellular level. Among various imaging methods, fluorescence imaging has prominent advantages [4,5,6]. After the introduction of fluorescent substances into biological cells or tissues, special excitation lights can be used to activate the substances to achieve safe and noninvasive detection [7,8,9,10,11,12,13].

The traditional fluorescence imaging method still suffers from the self-fluorescence signals in organisms, high cell damage, and poor penetration depth of the excitation light. Especially, the penetration depth of visible light is limited to a few millimeters, and the excitation light does not penetrate deeply into the target tissues [14,15]. These factors greatly limit the short-wavelength fluorescence probes in clinical applications [16]. To overcome these difficulties, current research on fluorescent probes is gradually focusing on long-wavelength probes, such as red and near-infrared (NIR) fluorescent probes. In addition to its deeper photon penetration, NIR fluorescence causes less optical damage and avoids the interference caused by biological self-fluorescence, which leads to a higher signal-to-noise ratio for detection [17,18]. Therefore, NIR fluorescence imaging is becoming one of the most promising analytical tools in the field of bioimaging [19,20,21].

In recent years, boron–dipyrromethene (BODIPY) dyes have been widely studied and applied as ideal fluorescent bioprobes [22,23] and exhibit excellent optical properties, such as high modifiability, excellent fluorescence quantum yield, small emission bandwidth, and satisfactory detection sensitivity [24,25,26,27,28,29]. To achieve the goal of NIR emission for detection or imaging, a common method is to extend the conjugation of the molecular structure. However, this may deteriorate its solubility and increase the difficulty in synthesizing BODIPY. The construction and modification of D-A structures on BODIPY have also been proven as efficient methods to achieve NIR emission, which not only red-shift the emission but also avoid the above-mentioned disadvantages [30,31,32,33,34,35,36,37,38,39,40,41]. Additionally, in view of the main differences between the internal aqueous system of living organisms and the inherent hydrophobicity of BODIPY, it is particularly important to improve the water solubility of BODIPY through various modification methods [42,43]. Polyethylene glycol (PEG) is widely used as a nontoxic and nonirritating water-soluble polymer. By connecting hydrophilic chains to hydrophobic BODIPY molecules, an amphiphilic structure can be produced that is also conducive to the self-assembly of amphiphilic molecules to form functional nanoparticles (NPs).

In this work, two BODIPY molecules (BDP-A and BDP-B) were designed and synthesized. As illustrated in Figure 1, the D-A structure was constructed by introducing aromatic substituents at the 1, 3, 5, and 7 positions of the BODIPY core structure, in order to red-shift its fluorescent emission band to the NIR region. For comparison, typical electron-donating diphenylamino groups were selected instead of *tert*-butyl groups in the D-A structure. Because of its low toxicity and excellent water solubility, a PEG chain was introduced in the neutral position of BODIPY to make the molecules amphiphilic, which can form amphiphilic NPs by self-assembly when dissolved in an aqueous solution (Figure 1). Furthermore, the structure–activity relationship between the functional group and the BODIPY core structure was also explored to lay the foundation for optical and bioimaging applications. These self-assembled BODIPY NPs, with high quantum yield and good biocompatibility, could be very promising candidates for NIR bioimaging. 

## 2. Results and Discussion

### 2.1. Synthesis and Characterization

The synthesis route of the target molecule BDP-A/B is shown in Figure 2. During the synthesis process, 2,4-substituted pyrrole and benzaldehyde with a PEG chain attached to *para*-position were initially synthesized. This selected and optimized synthetic procedure not only increases the solubility of the intermediates but also decreases the difficulty of synthesis and purification. Moreover, this molecular geometry minimizes the influence of hydrophilic groups on the luminescence properties of the BODIPY chromophore. The structures of the target molecules were characterized by ^1^H NMR and HRMS, although it was difficult to ionize and accurately obtain the HRMS data of BDP-A/B because of the introduction of the PEG chains. The final characterization results indicated that the target compounds were successfully synthesized (Appendix A). Both showed good solubility in different solvents.

### 2.2. Absorption and Fluorescence

The UV–Vis absorption spectra of BDP-A/B were measured in the concentration range of 1 × 10^−6^~1 × 10^−4^ mol L^−1^ in CH_2_Cl_2_. The optical data measured are listed in Table 1, and the absorption spectra are presented in Figure 1. Both compounds exhibited two distinct absorption bands, and their absorption maxima were 566 nm and 645 nm. BDP-A exhibited two sets of sharp absorption bands in the range of 350~650 nm, which were attributed to the ^1^π–π* transitions localized on the BODIPY aromatic structure. Compared with BDP-A, the absorption band of BDP-B in the range of 400~750 nm was much wider and significantly red-shifted. It was assumed that the introduction of the electron-donating groups caused intramolecular charge transfer (ICT) from the diphenylamino groups to the BODIPY core. Accordingly, the molecular modification by introducing NPh_2_ groups led to an obvious bathochromic shift of its absorption bands. In addition, the UV–Vis absorption spectra of the two compounds were measured in solvents with different polarities (DMF, CH_3_OH, THF, CH_2_Cl_2_, and toluene). Both BDP-A and BDP-B exhibited sharp absorption peaks and minor solvatochromic effects (Appendix A), which also confirmed that their absorptions were mainly dominated by ^1^π–π^*^ transitions of the conjugated structures.

As shown in Figure 1 and Table 1, the fluorescence spectra were measured, and the emission peaks of BDP-A and BDP-B were 605 nm and 729 nm, respectively. Both compounds exhibited high photoluminescence quantum yields in solution (93.2% for BDP-A and 62.1% for BDP-B). Similar to the results of UV–Vis absorption, BDP-B exhibited a red-shifted emission band with a larger Stokes shift (84 nm) than that of BDP-A (39 nm). The solvatochromic effect of BDP-A/B was also investigated in five solvents with different polarities (Appendix A). The presence of oxygen also had no noticeable effect on their emission spectra (Appendix A). It was noted that BDP-B showed a very obvious solvation effect, which was attributed to the introduction of diphenylamino groups. Therefore, the emission from BDP-A was predominantly assigned to the ^1^π–π* state, whereas BDP-B predominantly emitted from the ^1^π–π*/ICT state.

### 2.3. Theoretical Calculation

To investigate the distribution of the orbital electron clouds of the HOMO and LUMO of BDP-A/B, the conformational optimization of the two compounds was carried out with density functional theory (Appendix A). The geometry of the molecules at the ground state (S_0_ state) and the S_1_-excited state was optimized with the density functional theory (b3lyp/6-31G*) and time-dependent DFT (pbe1pbe/6-31G(d) TD) methods. The calculations were implemented without the attached PEG chain. From the obtained calculation results, the HOMO of BDP-A/B was mainly concentrated in the core structure of BODIPY, indicating the π-conjugate orbital crossover. In addition, the HOMO of BDP-B was distributed over the substituents due to the introduction of the strong electron-donating NPh_2_ groups. The LUMOs of BDP-A/B were almost all arranged on the core structure of BODIPY, but the orbital distributions of BDP-A and BDP-B were somewhat different because of the presence of electron-donating groups. The calculated energy gap value (ΔE_g_) was 2.55 eV for BDP-A and 2.05 eV for BDP-B, owing to the attachment of electron-donating diphenylamino groups, and the decreased energy gap value caused a significant red-shifted emission.

The fluorescence properties of BDP-A/B were further investigated from the energy gap between the S_0_ and S_1_ states, as shown in Figure 1c. The vertical excitations were calculated based on the optimized ground state geometry (S_0_), and the emission was calculated based on the optimized geometry of the excited state (S_1_). CT stands for conformation transformation. The dihedral angle between the BODIPY core and the diphenylamine motif was quite different in S_0_ geometry and S_1_ state minimum energy geometry. Compared with the geometry configuration of S_0_ state with a dihedral angle of 54°, the two components in the optimized S_1_ excited state were much more coplanar, showing a dihedral angle of 37°. The geometric relaxation during photoexcitation endowed a significant influence on the energy levels of molecular orbitals. Compared with the S_0_ state, the geometric relaxation of the S_1_ state possessed a much smaller energy gap between HOMO and LUMO. Geometric relaxation was the main reason for the large Stokes shift of BDP-B, which is also consistent with its Jablonski diagram of fluorescence description (Appendix A). 

### 2.4. Self-Assembly and Characterization

For the purpose of possible bioapplications, NPs for BDP-A/B were prepared. Because of its amphiphilic molecular design, BDP-A/B can be dissolved in water and form NPs by self-assembly. In addition, critical micelle concentration (CMC) is a key factor in self-assembly. The CMC values of BDP-A/B were estimated by the similar PEGylated nanomicelles reported in the previous literature [44], which were approximately 1.08 μM. The formation of sphere-like NPs was found with TEM (Figure 2b, Appendix A), and the formed NPs were stably dispersed in water with an appropriate hydration diameter (~90 nm). After continuous observation of BDP-A NPs for 72 h, the hydrodynamic size obtained by DLS test did not significantly change (Figure 2c, Appendix A). Zeta potential measurements yielded relatively high negative zeta potentials (−38.58 mV for BDP-A NPs and −45.21 mV for BDP-B NPs) in PBS buffer solution at pH 7.4, indicating strong electrostatic repulsion between the nanoparticles, thus preventing their agglomeration to improve the colloidal stability of the nanoparticles under these conditions. The emission properties of these NPs also were investigated and are listed in Table 1. Compared with BDP-A/B, the emission bands of NPs broadened and significantly red-shifted, but the absorption did not significantly change (Appendix A). Because of the intermolecular interaction caused by aggregation, the ICT progress was also greatly enhanced. In summary, the ^1^π–π*-dominated BDP-A/B showed weak emissions and low quantum yields, but the ICT-dominated NPs exhibited more red-shifted emissions.

Table 2 lists the photophysical properties of the BODIPY NPs reported in the literature, which suggests that the self-assembled BODIPY NPs in this work have obvious advantages. Notably, the high fluorescence efficiency of BDP-A NPs indicated that their emission performances were not severely quenched therein. Therefore, X-ray single-crystal diffraction analysis was performed. Single crystals were obtained from BDP-A with unattached PEG chains, and the crystallographic data are summarized in Appendix A (CCDC number: 2214676). The crystal structure diagrams of BDP-A show that the *meso*-methoxybenzene unit was almost orthogonal to the plane of the BODIPY core, forming a dihedral angle of 67.7°. Moreover, the 1, −7 *tert*-butylphenyl groups also formed a certain dihedral angle with the BODIPY core plane (63.8° and 65.4°, Appendix A). Indeed, both a perpendicular *meso*-methoxybenzene unit and two twisted 1, −7 *tert*-butylphenyl groups provided a sufficient steric hindrance. According to the molecular packing diagrams, the distance between two adjacent centroids of BDP-A core planes was 4.86 Å, which is much larger than the typical π–π stacking distance (3.4–3.6 Å) [45], indicating that the BDP-A molecules were not prone to aggregation. Therefore, the unique structural engineering of the BODIPY molecule significantly inhibited intermolecular aggregation, resulting in excellent performance even in water.

### 2.5. Biocompatibility Assessments of BDP-A/B NPs

The biocompatibility of nanomaterials is of great significance for their biomedical applications. In this work, HepG-2 cells were used to evaluate the biocompatibility of BDP-A/B NPs. As shown in Figure 3, no significant cytotoxicity was observed in BDP-A/B NPs after 24 h, even at 75 μg/mL. The interaction between blood components and BDP-A/B NPs was studied with an in vitro hemolysis test. With the concentration of BDP-A/B NPs ranging from 1 to 50 μg/mL, the hemolysis rate did not significantly change (Appendix A, Appendix A), indicating that BDP-A/B NPs can be applied in biomedical imaging.

### 2.6. Monitoring BDP-A/B NPs in Cells

To monitor the destiny of BDP-A/B NPs in living cells, HepG-2 cells were incubated with BDP-A/B NPs in PBS (pH = 7.4) and examined with confocal laser scanning microscopy (CLSM). As shown in Figure 3c,d, in the first 2 h, no NIR fluorescence was observed. As time went on, the NIR fluorescence became stronger. At 8 h, NIR fluorescence filled the cytoplasm. BDP-A/B NPs were mainly located in the cytoplasm, which was different from the blue nuclei stained with DAPI. These results suggest that BDP-A/B NPs can be taken up by HepG-2 cells and localized in the cytoplasm. 

### 2.7. In Vivo Fluorescence Imaging of Mice Tumors

For a comparative study of the photophysical properties, BDP-B NPs were used in vivo bioimaging. Four-week-old nude female mice were injected with 4T1.2 cells into the hind limb and were used as tumor models. After 7 days, the tumor model was used for optical imaging. The BDP-B NPs were injected into the mice via their tail vein, and the imaging effect of the material was tested at the first and sixth hour after injection. Figure 4 shows that the brightness of tumor tissues was significantly enhanced with increasing time. This indicates that BDP-B NPs accumulate at the tumor site over time. The fluorescence intensity of BDP-B NPs revealed the outline of the tumor tissue, indicating that BDP-B NPs can be effectively used for tumor imaging.

### 2.8. In Vivo Metabolism of BDP-B NPs

In order to study the metabolic pathways of the nanomaterials, two mice were separately dissected, as shown in Figure 5. The mice were dissected and analyzed by one hour after injection of the NPs. One hour after BDP-B NPs were injected into the mice, the fluorescence intensity was lower in the upper abdominal cavity due to the low accumulation of the materials, while the brightness was highest in the lower abdominal cavity, in the stomach and intestine. Accordingly, it could be deduced that although the BDP-B NPs were injected through the tail vein, the BDP-B NPs followed the bile into the stomach through the hepatobiliary metabolic pathway due to the blood circulation, and finally the drug was metabolized through the digestive system. After six hours, the fluorescence was mainly concentrated in the intestinal region, while the brightness of the stomach was greatly reduced. This phenomenon further validates how the BDP-B NPs were metabolized.

The organs of the two mice were then removed and examined separately; most were found to fluoresce with different intensities (Figure 6). Notably, the presence of fluorescence in the kidneys of the mice indicated that the material was metabolized by the urinary system in addition to the digestive system. We hypothesize that this phenomenon is due to the rupture of the NPs. After the formation of NPs in a low-concentration environment, small particles formed with a size of less than 10 nm, which could then be filtered by the glomerulus and metabolized by the urinary system. Moreover, six hours later after the injection, the intensity of the kidney was higher than that of one hour; therefore, our speculation is well confirmed by this phenomenon. Six hours after injection, the brightness of tumor tissue was also higher than that at one hour after injection, which is consistent with the results of in vivo tumor imaging, indicating that the material accumulates at the tumor site and achieves tumor imaging through the enhanced permeability and retention effect (EPR) over time. By comparing the brightness of the tumor tissue with that of the liver tissue, we found that the fluorescence intensity of the tumor was higher than that of the liver, although the volume of the tumor was much smaller, which indicates that the material has a very good passive targeting effect after NP formation and that its NIR emission characteristics can play an important role in tumor imaging.

## 3. Experimental Section

All chemicals were used as procured. The solvents used for the synthesis were single-distilled, and, for studies, high-purity solvents were used. The synthetic routes for BDP-A and BDP-B are shown in Figure 2. Synthetic methods for intermediates of BDP-A/B are shown in Appendix A. Characterization methods and other materials used are provided in ESI in detail.

mPEG-CHO: NaOH and mPEG2000 were added into a 250 mL reaction flask, then water (50 mL) and tetrahydrofuran (50 mL) were added to dissolve 4-toluenesulfonyl chloride. The reaction was carried out at 0 °C for 5 h. After the reaction, the reaction solution was poured into 100 mL of ice water and extracted with dichloromethane (50 mL × 2). The resulting organic phase was dried using Na_2_SO_4_; the solvent was removed under vacuum; and colorless, oily, liquid mPEG-OTs were obtained. To a mixture of mPEG-OTs, p-hydroxybenzaldehyde, and anhydrous potassium carbonate, anhydrous N, N-dimethylformamide (DMF) was added. The system was nitrogen displaced, and then the temperature was raised to 130 °C to reflux the reaction overnight. After the reaction solution was cooled to room temperature, the solvent was removed under vacuum, and dichloromethane (50 mL) was added. The dichloromethane solution was extracted using deionized water (50 mL × 2). The resulting organic phase was dried using Na_2_SO_4_, and then the solvent was removed under vacuum. The resulting crude product was purified by column chromatography V (DCM):V (CH_3_OH) = 10:1 to produce a white, solid mPEG-CHO. Yield: 43%. ^1^H NMR (400 MHz, CDCl_3_) δ 9.90 (*s*, 1H), 7.85 (*d*, *J* = 8.8 Hz, 2H), 7.04 (*d*, *J* = 8.7 Hz, 2H), 4.26–4.20 (m, 2H), 3.91 (*dd*, *J* = 5.4, 4.2 Hz, 2H), 3.69–3.63 (*m*, 163H), 3.40 (*s*, 3H).

BDP-A: A mixture of A3 and mPEG-CHO was dissolved in anhydrous dichloromethane (50 mL), and one drop of trifluoroacetic acid was added to initiate the reaction. The reaction system reacted in an ice bath under lightproof conditions for 12 h. Then, DDQ was added to the reaction system, and the reaction continued for 5 h. After the reaction, the reaction solution was extracted using a saturated aqueous sodium bicarbonate solution (50 mL × 3). The resulting organic phase was dried using Na_2_SO_4_, and then the solvent was removed under vacuum to produce a purple solid. The resulting solid was dissolved using 50 mL of dichloromethane, 0.8 mL of DIEA was added and stirred for 30 min, and then 1 mL of BF_3_·Et_2_O was slowly added to the reaction system, at which time the reaction solution gradually changed from purple to dark red. The reaction was carried out at room temperature for 5 h. After the reaction, the reaction solution was extracted with deionized water (50 mL × 2), the resulting organic phase was dried with Na_2_SO_4_, and then the solvent was removed under vacuum. The resulting crude product was purified by column chromatography V (DCM):V (CH_3_OH) = 10:1 to produce a purplish–red solid BDP-A. Yield: 26%. ^1^H NMR (400 MHz, CDCl_3_) δ 7.84 (*t*, *J* = 8.6 Hz, 4H), 7.43 (*dd*, *J* = 17.8, 6.8 Hz, 4H), 6.88 (*d*, *J* = 8.3 Hz, 4H), 6.69 (*dd*, *J* = 12.0, 8.5 Hz, 6H), 6.54 (s, 2H), 5.96 (*d*, *J* = 8.6 Hz, 2H), 3.69–3.55 (*m*, 172H), 3.38 (*s*, 3H), 1.35 (*s*, 18H), 1.15 (*d*, *J* = 26.3 Hz, 19H).

BDP-B: The mixture of B4 and mPEG-CHO was dissolved in anhydrous dichloromethane (50 mL), the reaction was initiated by adding a drop of trifluoroacetic acid, and the reaction system reacted in an ice bath under lightproof conditions for 12 h. Then, DDQ was added to the reaction system, and the reaction was continued for 5 h. After the reaction, the reaction solution was extracted with saturated aqueous sodium bicarbonate solution (50 mL × 3). The resulting organic phase was dried using Na_2_SO_4_, and then the solvent was removed under vacuum to obtain a dark-green solid. The obtained solid was dissolved using dichloromethane (50 mL), DIEA (0.8 mL) was added and stirred for 30 min; then, BF_3_·Et_2_O (1 mL) was slowly added to the reaction system, at which time the reaction solution gradually changed from dark green to blue-purple; and the reaction was performed at room temperature for 5 h. After the reaction, the reaction solution was extracted with deionized water (50 mL × 2), the resulting organic phase was dried using Na_2_SO_4_, and then the solvent was removed under vacuum. The obtained crude product was purified by column chromatography V (DCM):V (CH_3_OH) = 10:1 to obtain a dark-green solid BDP-B. Yield: 21%. ^1^H NMR (400 MHz, CD_2_Cl_2_) δ 7.84 (*d*, *J* = 8.8 Hz, 4H), 7.41–7.31 (*m*, 8H), 7.23 (*d*, *J* = 7.7 Hz, 8H), 7.16 (*t*, *J* = 7.3 Hz, 4H), 7.08 (*d*, *J* = 8.8 Hz, 4H), 7.01–6.89 (*m*, 6H), 6.81 (*dd*, *J* = 15.4, 7.6 Hz, 6H), 6.62 (*s*, 2H), 6.02 (*t*, *J* = 9.1 Hz, 2H), 3.65 (*d*, *J* = 6.0 Hz, 163H), 3.38 (*s*, 3H).

BDP-A/B NPs: BDP-A/B (1 mg) was separately dissolved in 20 mL of deionized water, treated with ultrasound for 30 min, and centrifuged for 5 min. The supernatant collected after centrifugation comprised BDP-A/B NPs.

## 4. Conclusions

In conclusion, two BODIPY molecules were successfully synthesized that can form NPs by self-assembly in water. Both BODIPY molecules showed red-to-NIR emission and large Stokes shifts that were well explained by theoretical calculations. They maintained not only good photophysical properties after forming NPs but also presented good biocompatibility. Both molecules were internalized by cells and had better imaging effects. In vivo imaging results of BDP-B NPs indicated that this nanomaterial can clearly show the shape of a tumor and be metabolized from the body through both liver and kidney pathways. Therefore, we think that self-assembled BODIPY nanoparticles are safe and effective NIR fluorescent materials for bioimaging.

## Data Availability

Not applicable.

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
