# Peer review of "Self-Assembled BODIPY Nanoparticles for Near-Infrared Fluorescence Bioimaging"

_molecules, 2023, doi:10.3390/molecules28072997_

Round 1

Reviewer 1 Report

I have gone through the article entitled "Self-assembled BODIPY Nanoparticles for Near-infrared Fluorescence Bioimaging".

This article seems interesting in terms of the synthesis and applications of two new dipyrromethene (BODIPY) molecules (BDP-A and BDP-B) and their nanoparticles in bioimaging.

The level of details and chemical information is well explained for the BDP-A and BDP-B. Also, the single-crystal X-ray structure is reported for BDP-A.

The photophysical and biocompatibility properties of the Nps were correctly explored.

Some minor points need to be addressed before the final decision:

1. ESI-HRMS data is missing in the manuscript.

2. The chemical structure for the different compounds has to be added in each NMR spectrum in the supporting information section.

 3. The 13C NMR spectra are missing in both the SI section and the manuscript.

4.  The DLS studies need to be correctly conducted and reported in this work.

5. Also, the zeta potentials for the different NPs have to be done and informed.

Author Response

  1. ESI-HRMS data is missing in the manuscript.

Response 1: Thanks for your nice comments. The HRMS data have been added in ESI file, which were highlighted in yellow.

  1. The chemical structure for the different compounds has to be added in each NMR spectrum in the supporting information section.

Response 2: Thanks for your suggestion, the chemical structure for the different compounds have been added in each spectrum in ESI.

  1. The 13C NMR spectra are missing in both the SI section and the manuscript.

Response 3: The 13C NMR spectra of A3 and B4 have been added in ESI which were highlighted in yellow.

4.The DLS studies need to be correctly conducted and reported in this work.

Response 4: The particle size distribution of BDP-A nanoparticles was obtained by DLS test, which was illustrated in Figure 1 below. As the absorption wavelength of BDP-B nanoparticles coincides with the measured wavelength of the instrument to a large extent, the measurement error was too large. Therefore, the particle size of BDP-B nanoparticles was counted and discussed by TEM images finally.

Figure 1. Size distribution of self-assembly of BDP-A in water determined by DLS.

  1. Also, the zeta potentials for the different NPs have to be done and informed.

Response 5: Thanks for your suggestion, the zeta potentials for the BDP-A/B NPs have been added in manuscript which were highlighted in yellow.

Reviewer 2 Report

According to the fundamental statement of the work “Thanks to the amphiphilic molecular design, BDP-A/B can dissolve in water and form NPs by self-assembly.”

But the formation, and hydrodynamic diameter of spherical NPs was only proven by TEM (Figure 2b, S14). In this way the existing of the self-assembled, associated molecules in aqueous phase should be proven by other experimental techniques ( for example IFT (10.1007/s11743-017-2025-x) or ITC (10.3390/nano11123288) method.

The hydrodynamic diameter of the self-assembled “particles” also should be determined by other independent methods like dynamic light scattering measurement.

Author Response

But the formation, and hydrodynamic diameter of spherical NPs was only proven by TEM (Figure 2b, S14). In this way the existing of the self-assembled, associated molecules in aqueous phase should be proven by other experimental techniques ( for example IFT (10.1007/s11743-017-2025-x) or ITC (10.3390/nano11123288) method.

Response: Thanks for your nice comments and suggestions. To demonstrate the formation of nanoparticles, we performed TEM characterization of BDP-A/B. Due to the limited experimental conditions, the critical micelle concentration (CMC) cannot be measured. Therefore, we referred to the literature (doi.org/10.1016/j.dyepig.2019.108105) which has similar molecular structure to BDP-A/B, and we estimate that the CMC value of BDP-A/B is similar to that of reported work. The literature was cited in the manuscript accordingly.

The hydrodynamic diameter of the self-assembled “particles” also should be determined by other independent methods like dynamic light scattering measurement.

Response: Thanks for your suggestion. The particle size distribution of BDP-A nanoparticles was obtained by DLS test. As the absorption wavelength of BDP-B nanoparticles coincides with the measured wavelength of the instrument to a large extent, the measurement error was too large. Therefore, the particle size of BDP-B nanoparticles was counted and discussed by TEM images finally.

Reviewer 3 Report

In this manuscript, Zhu and Liu reported the synthesis of new BPDIPY derivatives BDP-A and BDP-B with PEG chains to increase their water solubility, evaluated their fluorescence property in NIR region, and used them for fluorescence imaging of mice tumors. The molecular design for BDP-A/B appears to be working well in this study. I am not a specialist in bioimaging, but I believe the following minor points about the parts of synthesis and fluorescence properties need to be improved before this paper is accepted.

1) Characterization of the synthesized compounds appears to be inadequate. As far as I checked, B4, BDP-A, an BDP-B are novel compounds, but HRMS data for BDP-A and BDP-B are missing.

In addition, since HRMS data shown in Figure S3 and Figure S8 do not appear to match the theoretical isotope patterns, please add some explanation on this point or show theoretical MS pattern in SI to rationally explain these HRMS results. 

2) Purity of the products, especially BDP-A, appears to be low. Figure S4, 1H NMR of BDP-A, contains several small peaks around 5 ppm, 4 ppm and 2 ppm. Their minor byproducts may affect the fluorescence properties and biological activities. Therefore, it is necessary to further purify the products as much as possible. Or it would be necessary to rationally justify that these impurities do not affect the results of this study.

3) Table 1 shows optical properties of BDP-A, BDP-A NPs, BDP-B, and BDP-B NPs. The table caption says that they are analyzed in dichloromethane, but I think NPs are prepared in water. Please show the correct solvent conditions.

4) The authors discuss the result of DFT calculation, but do not provide any details on how they are calculated. Please show the details (e.g. TDDFT/B3LYP/6-31G(d)). 

5) Single-crystal X-ray structure of BDP-A with unattached PEG chains is shown. If the compound is a new compound and was synthesized in this study, please show the synthetic route and characterization data such as 1H NMR and HRMS. If the compound is known and has been synthesized in a previous paper, please cite that. 

In addition, please show more details of crystallographic values such as R1, wR2, and GOF in Table S1 to help readers understand the quality of the crystallographic analysis.

6) Based on the crystal structure, the authors discuss the possibility that π-π stacking aggregation does not occur in BDP-A NPs. I think this idea is reasonable, but one question arises. Why is the quantum yield of BDP-A NPs so low compared with BDP-A when there is no π-π aggregation? What factors decrease the quantum yield? Please add comments on these points, if possible.

Author Response

1) Characterization of the synthesized compounds appears to be inadequate. As far as I checked, B4, BDP-A, an BDP-B are novel compounds, but HRMS data for BDP-A and BDP-B are missing. In addition, since HRMS data shown in Figure S3 and Figure S8 do not appear to match the theoretical isotope patterns, please add some explanation on this point or show theoretical MS pattern in SI to rationally explain these HRMS results.

Response 1: Thanks for your suggestion. Although different methods have been tried to characterize BDP-A/B by HRMS, it is difficult to ionize and obtain accurately characterization data due to the introduction of PEG chains. In addition, the explanation on the HRMS results of A3 and B4 have been added in ESI which were highlighted in yellow.

2) Purity of the products, especially BDP-A, appears to be low. Figure S4, 1H NMR of BDP-A, contains several small peaks around 5 ppm, 4 ppm and 2 ppm. Their minor byproducts may affect the fluorescence properties and biological activities. Therefore, it is necessary to further purify the products as much as possible. Or it would be necessary to rationally justify that these impurities do not affect the results of this study.

Response 2: Sorry for the mistake. BDP-A was purified and re-characterized by 1H NMR, and the characterization results have been added to ESI. In addition, in the follow-up photophysical studies, no suspicious phenomenon was found. Therefore, we believe that these impurities had no influence on the experimental results, and the biological test results also confirmed our judgment.

3) Table 1 shows optical properties of BDP-A, BDP-A NPs, BDP-B, and BDP-B NPs. The table caption says that they are analyzed in dichloromethane, but I think NPs are prepared in water. Please show the correct solvent conditions.

Response 3: Sorry for the mistake. NPs are prepared in water indeed. We have corrected the table caption in the manuscript which are highlighted in yellow.

4) The authors discuss the result of DFT calculation, but do not provide any details on how they are calculated. Please show the details (e.g. TDDFT/B3LYP/6-31G(d)). 

Response 4: Thanks for your suggestion. The geometry of the molecules at the ground state (S0 state) and the S1 excited state was optimized by density functional theory (b3lyp/6-31G*) and time-dependent DFT (pbe1pbe/6-31G(d) TD) methods. The details of DFT and TDDFT have been added to the manuscript which are highlighted in yellow.

5) Single-crystal X-ray structure of BDP-A with unattached PEG chains is shown. If the compound is a new compound and was synthesized in this study, please show the synthetic route and characterization data such as 1H NMR and HRMS. If the compound is known and has been synthesized in a previous paper, please cite that. 

In addition, please show more details of crystallographic values such as R1, wR2, and GOF in Table S1 to help readers understand the quality of the crystallographic analysis.

Response 5: Thanks for your suggestion, the synthetic route, 1H NMR and crystallographic values mentioned have been added to the ESI and highlighted in yellow.

6) Based on the crystal structure, the authors discuss the possibility that π-π stacking aggregation does not occur in BDP-A NPs. I think this idea is reasonable, but one question arises. Why is the quantum yield of BDP-A NPs so low compared with BDP-A when there is no π-π aggregation? What factors decrease the quantum yield? Please add comments on these points, if possible.

Response 6: For BODIPY fluorescent dyes, their luminescence properties are easily affected by solvents, while water, as a typical polar solvent, will lead to a certain degree of quenching of BODIPY fluorescence. When BDP-A forms NPs in water, the polyethylene glycol in the outer layer of nanoparticles cannot completely block the influence of aqueous solution on BDP-A inside the particles. Still, the luminescence properties of BDP-A will be affected by aqueous solution, resulting in a fluorescence quenching.

Round 2

Reviewer 1 Report

All concerns have been answered, the paper can now be accepted for publication. 

Author Response

Response: Thanks for your nice comment.

Reviewer 2 Report

Taking into account "the limited experimental conditions", I suggest the  determination of  critical micelle concentration (CMC) by a robust method using coumarin-6 as a fluorescent probe (DOI: 10.1039/C9AY00577C ). Or find a relevant data from literature isted of cited article (doi.org/10.1016/j.dyepig.2019.108105) which does not give any informaion about the cmc.

Author Response

Response: Sorry for the mistake. We cited another article from the same research group which has similar molecular structure, thus the article does not give any information about the cmc. Now we corrected the citation, and the figure and data form the reference are provided below. Accordingly, we estimate that the CMC values of BDP-A/B is similar to that of the PEGylated nano-micelle (doi.org/10.1016/j.dyepig.2020.108651), which are about 1.08 μM. We have also added the discussion in the manuscript and highlighted it in red.

Yong Deng, Fei Huang, Jing Zhang, Jie Liu, Bing Li, Ruizhuo Ouyang, Yuqing Miao, Yun Sun*, Yuhao Li*. PEGylated iridium-based nano-micelle: Self-assembly, selective tumor fluorescence imaging and photodynamic therapy. Dyes and Pigments 182 (2020) 108651. (doi.org/10.1016/j.dyepig.2020.108651)

Reviewer 3 Report

The authors seem to have addressed most of the points raised, but there is one area that I would like to see revised. In the section 2.1 "Synthesis and characterization" or the section 3 "Experimental section",  please add a comment similar to the following sentence (the bold part) about HRMS analysis of the target compounds (BDP-A/B).

"The structures of the target molecules were characterized by 1H NMR and HRMS, although it was difficult to ionize and obtain accurately HRMS data of BDP-A/B due to the introduction of PEG chains."

In addition, please add a comment on 13C NMR analysis on the first page of the supporting information.

Author Response

The authors seem to have addressed most of the points raised, but there is one area that I would like to see revised. In the section 2.1 "Synthesis and characterization" or the section 3 "Experimental section", please add a comment similar to the following sentence (the bold part) about HRMS analysis of the target compounds (BDP-A/B).

Response: Thanks for your nice comments and suggestion. the comment has been added in the section 2.1 which are highlighted in red.

In addition, please add a comment on 13C NMR analysis on the first page of the supporting information.

Response: The comment on 13C NMR have been added on the first page of ESI which are highlighted in red.